# Influence of the Substituent’s Size in the Phosphinate Group on the Conformational Possibilities of Ferrocenylbisphosphinic Acids in the Design of Coordination Polymers and Metal–Organic Frameworks

**DOI:** 10.3390/ijms241814087

**Published:** 2023-09-14

**Authors:** Ruslan P. Shekurov, Mikhail N. Khrizanforov, Ilya A. Bezkishko, Kamil A. Ivshin, Almaz A. Zagidullin, Anna A. Lazareva, Olga N. Kataeva, Vasili A. Miluykov

**Affiliations:** 1Arbuzov Institute of Organic and Physical Chemistry, FRC Kazan Scientific Center of RAS, 420088 Kazan, Russia; shekurovruslan@gmail.com (R.P.S.); bezkishko@gmail.com (I.A.B.); kamil.ivshin@yandex.ru (K.A.I.); almaz_zagidullin@mail.ru (A.A.Z.); aal.lazareva11@mail.ru (A.A.L.); olga-kataeva@yandex.ru (O.N.K.); miluykov@iopc.ru (V.A.M.); 2A.M. Butlerov Chemistry Institute, Kazan Federal University, 420008 Kazan, Russia; 3Institute of Fundamental Medicine and Biology, Kazan Federal University, 420008 Kazan, Russia

**Keywords:** metal–organic frameworks, coordination polymers, phosphinates, ferrocene derivatives, X-ray

## Abstract

This paper illustrates how the size and type of substituent *R* in the phosphinate group of ferrocenyl bisphosphinic acids can affect conformational possibilities and coordination packing. It also demonstrates that *H*-phosphinate plays a key role in variational mobility, while Me- or Ph- substituents of the phosphinate group can only lead to 0D complexes or 1D coordination polymer. Overall, this paper provides valuable insights into the design and construction of coordination polymers based on ferrocene-contained linkers. It sheds light on how different reaction conditions and substituents can affect conformational possibilities and coordination packing, which could have significant implications for developing new polymers with unique properties.

## 1. Introduction

Recently, metal–organic frameworks (MOFs) have garnered widespread research attention in many fields, including gas separation, energy storage and conversion, wastewater treatment, catalysis, sensors and others [1,2,3,4,5,6,7,8,9,10]. The design and synthesis of coordination polymers and MOFs using ferrocene-containing ligands have emerged as a crucial area of research in coordination chemistry [11,12,13,14,15,16,17,18]. Ferrocene-based MOFs not only maintain the intrinsic porous frameworks of MOFs, but also exhibit high stability, outstanding reversible redox property and extraordinary dual optical- and electro-chemical activities due to the presence of the ferrocenyl group. These coordination polymers have shown a range of properties, including redox activity [17], luminescent [19], magnetic [20], sorption properties [21] and other features [22].

While ferrocenes are renowned for their synthetic versatility, the coordination properties of ferrocene-based carboxylic acids have been most extensively studied [23,24,25,26,27]. The conformational flexibility of the cyclopentadienyl (Cp) rings allows for structural variety in coordination polymers that can be controlled by substituent choice. Meanwhile, phosphorus acid derivatives cannot have the conjugation between aromatic moiety and phosphorus group that can lead to additional free rotation and additional flexibility of ligand (Figure 1) [28]. Also, a versatile phosphorus chemistry and additional R-substituent offers great opportunities for efficient fine-tuning of donor–acceptor properties of ferrocene fragment from a fundamental point of view. For comparison, carboxylic acid-based linkers can only be modified by changing the spacer group between the carboxylic acids, whereas phosphinic acids can be additionally modified by either alkyl or aryl groups at the phosphorus atom.

This possibility of tuning electronic and steric properties with two aryl/alkyl substituents directly attached to phosphorus atom of phosphinic acids translates into unique properties of materials that cannot be replicated by the utilization of analogous phosphonates or carboxylates [29,30,31]. The properties of phosphinates fall in many ways between the properties of carboxylates and phosphonates, combining somewhat stronger binding to hard metal ions compared to carboxylates with less damaging effect on surfaces and more predictable coordination modes than phosphonates. The phosphinate group also falls between carboxylates and phosphonates in terms of Pearson’s theory of hard and soft acids and bases [31]. The acidity of phosphinic acids also lies between the acidity of carboxylic and phosphonic acids [32].

In acidic environment, polytopic carboxylates MOFs, frequently displaying porosity, often composed of discrete oxometalate secondary building units are bound together by polytopic carboxylate linkers [33]. In the case of phosphinates, the tendency to create layered structures is suppressed, whereas M-O-P-O-M polymeric chains are common coordinating motifs [34,35,36,37]. Phosphinate MOFs in particular provide an alternative to well-known carboxylate MOFs, with the advantage of stronger bond with trivalent metal centers, which results in increased MOF hydrothermal stability [38,39,40,41].

In general, the use of bulky substituents in phosphinate groups can limit the free rotation of the molecule and affect the coordination packing mode in crystals. This is because larger substituents may not allow for complete rotation of the phosphinate group, which can impact on the formation of a three-dimensional supramolecular structure. However, it is important to note that this effect may vary depending on the specific substituent used and other factors such as reaction conditions and solvent. Therefore, careful consideration must be given to these factors when designing and synthesizing coordination polymers and MOFs with specific structural and functional properties [31].

The investigation of the flexibility of bisphosphinic acids was initiated by Midollini et al. This research has contributed to our understanding of the structural properties and behavior of coordination polymers based on alkyl spacers between phosphinate groups. These polymers typically involve divalent metal ions bridged by phosphinate oxygen atoms, and can exhibit a range of structural arrangements, from one-dimensional to three-dimensional architectures. The specific structure formed depends on various factors, such as the nature of the metal used and the reaction conditions employed. By studying these factors, researchers can gain insights into how to design coordination polymers with specific structural and functional properties for various applications [42,43].

The ethylene chain’s ability to adopt different conformations enables the formation of porous coordination polymers that exhibit reversible crystal transformation through hydrogen bonding interactions. This property is particularly useful in the design and synthesis of materials for gas storage, separation, and catalysis. Overall, this property highlights the importance of understanding the conformational flexibility of ligands in coordination polymer synthesis and design [44]. In recent studies conducted by a scientific group, it was demonstrated that incorporating an additional spacer between the alkyl groups in the skeleton of a coordination polymer can increase rigidity and distance between the two phosphinic groups. This structural modification can prevent the P(xyl)P^2−^ anion ligand from chelating a single metal center, due to the increased distance between the two phosphinic groups. This finding has important implications for designing and synthesizing coordination polymers with specific structural and functional properties, particularly in applications where metal chelation is undesirable or needs to be avoided [45]. According to the information provided in research, the elongation of the chain connecting two diphosphinate moieties can promote the formation of stable two-dimensional (2D) networks in coordination polymers. Additionally, the presence of a xylyl spacer between these moieties can block the conformation of the hybrid polymer through π-π interactions between the xylyl and phenyl groups. This effect can favor the formation of a three-dimensional (3D) crystalline anhydrous network instead. These findings highlight how subtle changes in ligand structure can impact the formation and properties of coordination polymers, and underscore the importance of careful design and synthesis in this field. The ferrocenylen and alkylen spacers are examples of such ligands (a balance between rigidity and flexibility), as they exhibit good flexibility due to their ability to turn Cp rings 360° around the axis of ferrocene. However, they also have a rigid organometallic ferrocene skeleton that provides additional advantages for ferrocene-based ligands. These findings highlight the importance of balancing different structural properties when designing and synthesizing coordination polymers with specific properties and functions.

The ferrocenylphosphinic acids have one more flexible part in ligand: free rotation of the phosphinate group around P-Cp bond (Figure 1). This property can enable more controlled design of MOF formation, as demonstrated in a previous article. Recently, a convenient method for preparing mono- and bisferrocenylphosphinic acids [46] has been developed by our group (Figure 2). The paper also presents an interesting dependency of coordination packing on the substituent of phosphorus atom and reaction conditions of bisferrocenylphosphinates. These findings highlight the importance of understanding ligand structure and its impact on coordination polymer formation and properties.

## 2. Results and Discussion

### 2.1. Supramolecular Architecture of Diammonium Ferrocene-1,1′-diyldi(R-phosphinate)

The effect of substituents in the ferrocene fragment on the conformational behavior and strength of hydrogen bonds is an important issue in coordination chemistry. Ferrocenylphosphinic acids **1**–**3** are particularly interesting in this regard, as they offer promising opportunities for designing supramolecular systems. Unlike carboxylic acids, ferrocenylphosphinic acids do not have conjugation with the ligand framework, making them more labile and easier to manipulate. Additionally, their donor–acceptor properties can be finely controlled by varying the fourth R-substituent at the P-atom. These findings demonstrate the potential of ferrocenylphosphinic acids as versatile tectons for designing coordination polymers with specific properties and functions.

In the case of ammonium salts [47,48], the formation of supramolecular coordination structures due to the formation of hydrogen bonds between ammonium and phosphinates was observed (Figure 3). These systems are promising to study as highly soluble candidates for ARFB (aqueous redox flow battery) applications [49]. The ability to form supramolecular structures with high solubility is a desirable property for ARFBs, as it can improve their performance and efficiency. By understanding the factors that influence the formation of these structures, researchers can design and optimize new ferrocene-based materials for ARFB applications with improved properties and performance [50].

Both *H*-phosphinates ((NH_4_)_2_fcd*H*p) **1a** and *Me*-phosphinates ((NH_4_)_2_fcd*Me*p) **2a** are characterized by the formation of 3D supramolecular structures. These structures are stabilized through a branched system of hydrogen bonds between phosphinate groups and ammonium cations, resulting in a complex 3D supramolecular architecture. The diagram above illustrates this structure, where each phosphinate group is bonded to four ammonium cations, and each ammonium cation is bonded to four phosphinate anions. This type of supramolecular structure has important implications for various fields, including materials science, chemistry, and biology (Figure 1).

The small size of the proton enables the phosphinate group to adopt a favorable conformation for forming hydrogen bonds with ammonium cations. In the case of 1,1′-ferrocenylene-bis(methylphosphinic) acid **2** it was observed that the methyl substituent in the salt does not hinder the formation of hydrogen bonds in 3D space (Figure 1, on the right). This finding is significant as it demonstrates that even with a bulkier than proton substituent, hydrogen bonding can still occur and lead to supramolecular structures. While the larger size of the methyl group cannot allow for complete rotation of the phosphinate group, it does not prevent the formation of a three-dimensional supramolecular structure. However, in the case of bulky phenyl substituent at phosphorus atom (*Ph*-phosphinates, (NH_4_)_2_fcd*Ph*p **3a**), the packing density in the salt crystal is significantly reduced under the same conditions, despite the presence of solvate water molecules in the crystal (Figure 2). In this series of compounds, (NH_4_)_2_fcd*Ph*p **3a** stands out because the independent part of the elementary cell consists of one molecule of fcd*Ph*p and two independent NH_4_^+^ cations. In the formation of a hydrogen-bonded chain, only one cation (highlighted in blue in Figure 2) participates, while the second cation (magenta) is bounded to water molecules and does not participate in the formation of a polymeric chain. Due to the asymmetric structure of the anion, there is almost unrestricted internal rotation around the P–Cp bond. The phosphinate group near the P(2) atom is positioned in a manner where one of the P(2)–O bonds is obscured by the C–C bond of the cyclopentadienyl ring, and the phenyl group is perpendicular to its plane. Another phosphinate group has a distinct orientation: the cyclopentadienyl ring is located in the middle of the OPO fragment, and the relative arrangement of the P–C(Ph) bond and C–C bond of the cyclopentadienyl ring is similar to that of shaded. Therefore, the existence of numerous proton contributors and receptors in diammoniumferrocene 1,1′-diyldi(*R*-phosphinates) crystals, along with the adaptable conformation of their anions, establishes a complex system of branched intermolecular hydrogen bonds. The size of the resulting supramolecular assemblies is influenced by the substituents in the phosphorus atom.

### 2.2. Crystal Structures and Control of Coordination Mode with M^2+^ and M^3+^

Similar conformational flexibility is observed in coordination polymers, analogous to ammonium salts. H_2_fcd*H*p **1** exhibits a wide range of coordination possibilities depending on the metal or solvent, which will be discussed below. However, H_2_fcd*Me*p **2** and H_2_fcd*Ph*p **3** demonstrate less flexible conformational possibilities and consequently result in a narrower range of coordination compounds. The explanation for this phenomenon will be discussed in this paper.

A solvothermal reaction is carried out at 80 °C for 12 h between M^2+^ (M = Zn or Co) and ferrocene-based ligand H2fcd*H*p **1** in a mixed DMF/MeOH solvent, resulting in the formation of orange crystals of **4a** (for M = Zn, Figure 3) or dark blue crystals of **4b** (for M = Co) [51]. The metal center in polymers **4a**,**b** is tetrahedrally coordinated by four O atoms from four phosphinate groups of two ligands, with bond lengths ranging from 1.926 to 1.936 Å for **4a**. The coordination of ligands to the metal ions resulted in the construction of a one-dimensional (1D) helical chain structure. The eclipse conformation of the cyclopentadienyl rings of ferrocene and coordination of two metal ions leads to tension between planes of Cp rings (dihedral angle equal 9.11°).

To inspect the role of solvent on the formation type of coordination polymer, the DMF/MeOH solvent pair was employed instead of H_2_O. In room temperature, 2D-layered structures of **5** of M^2+^ (M = Cd [52] (**a**), Co [53] (**b**) or Mn [54] (**c**)) with octahedral geometry of metal center are formed, which consist of two molecules of water in axial positions and four oxygen atoms of four ligands. Each MO_6_ polyhedron is further linked to another MO_6_ fragment through two bridging phosphinate groups, so that an infinite chain of metal centers results, the nearest M–M distance being 5.526 Å for **5a**, 5.394 Å for **5b**, 5.506 Å for **5c**. *Trans*-orientation of Cp rings leads to the binding of phosphinate chains in 2D structure. Another type of polymer is formed due to the free rotation of phosphinate groups around P-Cp bond. The significance of the existence of water molecules in the lattice (Figure 4), both for crystallization and coordination, is widely acknowledged. This is because the propagation of structures in 2D extended networks can be attributed to strong hydrogen bonding interactions.

By utilizing more bulky substitutes on the phosphorus atom (R = Me) and a mixed DMF/MeOH solvent, two series of isoreticular 1D coordination polymers Mfcd*Me*p (**6**) were obtained, with M = Zn(II) (**a**), Co(II) (**b**), and Mn(II) (**c**). The compound **6c** was characterized via X-ray structural analysis. The structural identity of the **6a**, **6b**, and **6c** coordination polymers was validated through infrared spectroscopy and thermal gravimetric measurements (Appendix A).

Compound **6c** is a 1D coordination polymer that runs in the *0a* direction and bears similarity to the coordination polymers described for polymers **4**. The structure of **6c** consists of Mn ions that belong to two eight-membered cycles, which are arranged as Mn-O-P-O-Mn-O-P-O. The phosphinate group links two neighboring cycles, while the ferrocene molecule exhibits a slightly twisted conformation with a very small shift in the cyclopentadienyl rings. The manganese ions are coordinated by four oxygen atoms from symmetrically dependent phosphinate groups, displaying tetrahedral coordination geometry. The metal polyhedrons MO_4_ of coordination polymers **6** exhibit a tetrahedral environment similar to **4** (Figure 5). However, the main difference in coordination packing between **6** and **4** lies in the conformation of the ferrocene moiety. The phosphinate groups’ gauche position leads to the realization of a chelate mode with two oxygen atoms of two phosphinates by one ligand towards the metal center. The other two oxygen atoms connect ions of metals and form a zigzag chain, so one diphosphinate connects three different metal centers, in contrast to **4** where each ligand connects only two metal ions in a helical chain structure. The distance between ions Mn(II) in **6c** is 4.365 Å, while in **4a** it is only 4.408 Å.

When we changed the solvent conditions in synthesis from the mixed DMF/MeOH solvent to H_2_O, we expected to obtain 2D coordination polymers similar to **5**. However, we obtained only crystals of polymers **6** with good yields and quality. To determine the reason for this different coordination behavior of ligands H_2_fcd*H*p **1** vs. H_2_fcd*Me*p **2**, we considered the main difference between them and coordination polymers **4** and **6** from **5**. In water solution, H_2_fcd*H*p **1** forms coordination polymers with an open shape of hydrophilic parts of the structure (Figure 6). Hydrophobic ferrocene moieties alternate with metal centers, phosphinic groups, and coordinate crystal water molecules. In contrast to **5**, the packing modes of **6** or **4** have a close shape of hydrophilic parts of the structure around the channel of metal centers and phosphinic groups located in a hydrophobic shell consisting of ferrocene moieties and the fourth substitute of the phosphorus atom, which is methyl in case of **6**. However, this type of packing mode is not suitable for **4** because *H* is not bulky. Nature has realized an excellent solution with a helical catena coordination type of polymer.

The main conformation difference of H_2_fcd*H*p in **4** and **5** is *cis*- and *trans*-orientation Cp rings of ferrocene moiety and positions of the oxygen atoms with relative to ferrocene sandwich into and outside for **4** and **5**, respectively.

In case of **5**, the outside conformation of phosphinate groups leads to an open access for the realization of a larger number of hydrogen bonds. However, apparently, more bulky methyl substitutes cannot realize outside the position of both oxygen atoms of the phosphinate group. This similar behavior was also observed in the crystal packing of ammonium salts of H_2_fcd*H*p **1a** and H_2_fcd*Me*p **2a** or H_2_fcd*Ph*p **3a**.

In the presence of a phenyl substituent at the phosphorus atom, various attempts with *d*- and *f*-metals under different conditions did not lead to the formation of coordination polymers suitable for identification. However, Fe^3+^ reacting with **3** leads to the formation of a tris-chelate discrete complex **7** (Figure 4). It was found that heating a mixture of 1,1′-ferrocenediyl-bis(phenylphosphinic acid) **3** and iron(III) chloride in DMSO or DMF at 150 °C leads to the formation of a tris-chelate complex **7**.

The structure of complex **7** was confirmed via X-ray structural analysis (Figure 7). It is interesting to note that proceeding of this reaction in coordinating solvents allowed us to synthesize four different polymorphs containing DMF, DMSO, THF, and methanol as the solvating solvent [55,56,57,58].

The central iron ion has an octahedral environment with practically equal distances of d(O-Fe) = 1.968–1.980 Å. The complex **7** realizes a monodentate chelating type of metal-ligand binding due to the *gauche*-conformation of the cyclopentadienyl rings of the monoanionic acid. The remaining mobile proton of the phosphinate group forms a strong intramolecular hydrogen bond with the P=O phosphinate group of the neighboring ligand.

It is interesting to compare the packaging of **1** with trivalent metal. In the case of iron (III) and aluminum (III), we were unable to obtain crystals, only aerogels of Fe(III) [53] and Al(III) [59] were formed. However, in the case of lanthanides, a crystalline structure of a 2D polymer was obtained. It has been established that heating a mixture of acid **1** and Sm(III) chloride in water up to 100 °C leads to the formation of a layered two-dimensional coordination polymer **8a** (Appendix A) [60].

According to the data of X-ray structural analysis, compound **8a** contains two different types of ligands in eclipsed conformations characterized by torsion angles of 71° and 75.4°, respectively. One type of ligand binds two samarium ions, forming an eight-membered cycle, while the other participates in stitching such cycles together through a bridging OPO bond, forming a two-dimensional structure (Figure 8).

The samarium ion has an octahedral environment, which is atypical for complexes obtained in an aqueous medium, and consists of six oxygen atoms from four phosphinate ligands. Two of the phosphinate ligands are chelating, while the other two are bridging, forming a polymeric chain.

It should be noted that the similarity in the metal coordination type in the series of 1,1′-ferrocenediyl-bisphosphinates is reflected in the equivalence of the angles of rotation of the cyclopentadienyl rings in tris-chelate complex **7** and coordination polymer **8a** (Figure 9).

The smaller size of the substituent at the phosphorus atom in 1,1′-ferrocenediyl-bis(*H*-phosphinic acid) (**1**) compared to 1,1′-ferrocenediyl-bis(phenylphosphinic acid) (**3**) leads to the free rotation of the phosphinate groups around the P-C bond, resulting in a bridging type of coordination for **1** and the formation of a coordination polymer **8a**. It should be noted that similar coordination polymers based on acid **1** are also formed for Eu(III), Dy(III), and Y(III) derivatives [61,62].

Thus, the size of the substituent at the phosphorus atom has a significant influence on the formation of structures with different architectures. The bridging type of coordination is observed for 1,1′-ferrocenediyl-bis(*H*-phosphinic acid) (**1**) due to the free rotation of the phosphinate groups around the P-Cp bond. The similarity in metal coordination type is reflected in the equivalence of angles of rotation of cyclopentadienyl rings in tris-chelate complex **7** and coordination polymer **8a**.

### 2.3. Role of 4,4′-Bipyridine in the Solvothermal Synthesis of MOFs Based on Ferrocenyl-R-phosphinates

It was interesting to study the behavior of ferrocenylphosphinates **1**–**3** and the role of substituents in the presence of an additional neutral linker (Appendix A). Three-dimensional coordination polymers based on acid **1** were obtained via complexing with nickel or cobalt nitrate. For example, the formation of 3D coordination polymer **9a** occurs when an equimolar mixture of acid **1**, 4,4′-bipyridine (bpy) and nickel nitrate is maintained in a mixture of methanol-DMF-water at a ratio of 8:2:1 for 12 h at 80 °C [63]. The structure of this polymer **9a** was confirmed via X-ray structural analysis (Figure 10). The formation of similar coordination polymers also occurs when replacing nickel nitrate (II) with cobalt nitrate (II).

The Ni atom in compound **9a** has an octahedral environment and is coordinated by four oxygen atoms from different phosphinate groups with almost identical Ni-O distances of 2.079 Å and 2.078 Å, and axially bounded to nitrogen atoms of bipyridyl with Ni-N distances of 2.095 Å and 2.115 Å. Each phosphinate ligand binds two nickel ions, forming an infinite chain consisting of eight-membered cycles. The 4,4′-bipyridyl molecules stitch the chains together in a two-dimensional direction, and due to the eclipsed conformation of the cyclopentadienyl rings of the ligand, which are almost in a *trans*-position (torsion angle equals 149.22°), the layers are stitched together into a 3D framework structure.

The space between the ferrocene fragments in the coordination polymer are filled with co-crystallized methanol molecules that are not connected by hydrogen bonds (Figure 11). Thus, the first representative of 3D porous coordination polymers based on ferrocene-containing phosphinic acids was obtained.

However, the analysis of MOFs **9a**,**b** via thermogravimetry showed that methanol molecules begin to leave the pores only at a temperature of 250 °C with simultaneous destruction of the polymer structure, which is reflected in the absence of a “plateau” on the curve (Appendix A).

Preliminary drying of this polymer at 150 °C under normal pressure or 250 °C under vacuum does not lead to the formation of free pores, as evidenced by the low nitrogen sorption (6 cm^3^/g) of compound **9a**. Presumably, methanol molecules are tightly located in the channels of the polymer and leave it only when the crystalline structure of the polymer is destroyed. Therefore, compound **9** can be classified as a first-type porous coordination polymer according to Kitagawa’s classification, [64] which is characterized by destruction of the crystalline structure upon the removal of guest molecules.

Thus, three-dimensional coordination polymers based on 1,1′-ferrocenediyl-bisphosphinic acid **1** and transition metal compounds can be obtained using additional ligands, in particular 4,4′-bipyridine. It is interesting that in the case of acids **2** and **3**, coordination polymers with 4,4′-bipyridine are not formed. In the case of **2** with Mn(II), even the presence of 4,4′-bipyridine leads to a polymer of type **6c**, while in the case of Ni(II), polymer **10** (Figure 12) is formed, where the ferrocene fragment is not even included in the coordination chain of the polymer and acts as a counterion in the nickel-4,4′-bipyridine chain. The similar compounds contain the phosphonate anion as the counterion to charge balance the cationic charge originating from the metal cation was observed in paper [65].

The structure of polymer **10** is composed of parallel straight chains of Ni-complex and ferrocene molecules linked by hydrogen bonds between oxygen atom of phosphinate group and water molecules. Both chains run in the [010] direction (Figure 12).

Each Ni ion is located at a crystallographic 2-fold axis, coordinated by four H_2_O molecules and two bridging 4,4′-bipyridine ligands forming a square bipyramidal coordination of the metal. The angle between pyridine planes is equal to 23.37(10)°, due to the different intermolecular interactions, namely the N1-pyridine forms hydrogen bonds with the oxygen atoms of phosphinate group of the ferrocene, while C–H···π contacts is observed for N2-pyridine. The intermolecular bonds are illustrated in Figure 13, and the parameters of the bonds are listed in Appendix A.

### 2.4. Physico-Chemical Properties

Our findings reveal the potential of 1,1′-ferrocenylenbis(*R*-phosphinic) acids to form not only 1D and 2D non-porous coordination polymers but also layered and 3D MOFs. Powder diffraction patterns of coordination polymers based on M(II) cations are presented in Appendix A. These synthesized polymers are primarily used in various electrochemical applications (as summarized in Table 1). This is mainly due to the presence of active redox pairs, primarily the reversible Fe^II^/Fe^III^ pair of ferrocene core. Regarding sorption properties, most polymers based on *R*-phosphinic acids are not suitable for such applications. However, an exception can be noted in the case of the 2D polymer **5c**, which exhibits exceptional selectivity towards water sorption, a characteristic attributed to the inherent structure of the polymer itself (for further details refer to the Appendix A). In terms of nitrogen sorption, only aerogels based on Fe^3+^ and Al^3+^ showed high values based on the sorption curves (Table 1).

Previously undescribed 1D coordination polymers **6a**–**c** were characterized by cyclic voltammetry. The redox properties of coordination polymers based on Mn(II), Zn(II), and Co(II) were investigated using a carbon paste electrode with a phosphonium salt serving as the binder, which has proven to be effective for these purposes. Among the metals, the oxidation potential decreases in the order Mn(II) > Co(II) > Zn(II) (Figure 14).

According to the data presented (Table 2), the oxidation of the ferrocene fragment in coordination polymers occurs at more positive potentials compared to pure ferrocene (0.45 V and ΔE_p_ = (E_p_^a^ − E_p_^c^) = 60 mV), similar to other ferrocenylphosphinates. However, when comparing 1D coordination polymers, there is an increase in ΔE_p_ when H is replaced with Me in the phosphorus atom. The oxidation peaks correspond to the oxidative transition from Fe(II) to Fe(III). This indicates that when the oxidation state of iron in the polymer changes, the contribution of the cyclopentadienyl rings toward stabilizing the oxidized form also changes.

## 3. Materials and Methods

### 3.1. General

The thermogravimetric analysis (TGA) was carried out on air using an Analysis Netzsch STA 409 PC Luxx (Netzsch-Gerätebau GmbH, Selb, Germany), with a heating rate of 5°/min in the temperature region from 20 up to 1200 °C. IR spectra of solid compounds have been registered using Vector-27 FTIR spectrometer (Bruker, Ettlingen, Germany) in the 400–4000 cm^−1^ range (optical resolution 4 cm^−1^); the samples were prepared as nujol mulls. Powder X-ray diffraction data were collected on a STOE Stadi P diffractometer (STOE & Cie GmbH, Darmstadt, Germany) with Cu-Kα1 radiation (λ = 1.5405 Å). CCDC 2258503 and 2258504 contain the supplementary crystallographic data for this paper.

### 3.2. X-ray Diffraction Analysis

Data sets for single crystals were collected on a Bruker Kappa APEX Duo diffractometer with graphite-monochromated Cu Kα radiation (λ = 1.54178 Å) at 150(2) K for **6c** and on a Bruker Kappa APEX diffractometer with graphite-monochromated Mo Kα radiation (λ = 0.71073 Å) at 150(2) K for **10**. Programs used: data collection APEX [66], data reduction SAINT [67], multi-scan absorption correction SADABS [68], structure solution SHELXT [69], structure refinement via full-matrix least-squares against F2 using SHELXL [69] Hydrogen atoms at phosphorus atoms and oxygen atom were revealed from difference Fourier map and refined isotropically.

Crystal data for **6c**: formula C_12_H_14_FeMnO_4_P_2_, M = 394.96 g/mol, triclinic, space group *P*-1 (No. 2), Z = 2, a = 8.5711(4) Å, b = 9.5437(5) Å, c = 10.4813(5) Å, α = 66.729(2)°, β = 75.136(2)°, γ = 72.702(2)°, V = 742.49(6) Å^3^, ρ_calc_ = 1.767 g·cm^–3^, μ = 16.879 mm^–1^, 16,951 reflections collected (−9 ≤ *h* ≤ 10, −11 ≤ *k* ≤ 11, −12 ≤ *l* ≤ 12), θ range = 4.651° to 70.267°, 2759 independent (R_int_ = 0.0351) and 2664 observed reflections [I ≥ 2σ(I)], 184 refined parameters, R(F) = 0.0268, wR(F^2^) = 0.0718, max (min) residual electron density 0.389 (−0.464) e Å^–3^.

Crystal data for **10**: formula C_10_H_16_N_2_NiO_4_·C_12_H_14_FeO_4_P_2_·2(H_2_O), M = 663.01 g/mol, monoclinic, space group *P*2/*c* (No. 13), Z = 2, a = 8.8000(18) Å, b = 11.320(2) Å, c = 14.480(3) Å, β = 106.20(3)°, V = 1385.2(5) Å^3^, ρ_calc_ = 1.590 g·cm^–3^, μ = 1.374 mm^–1^, 17,736 reflections collected (−11 ≤ *h* ≤ 11, −14 ≤ *k* ≤ 14, −15 ≤ *l* ≤ 15), θ range = 1.799° to 27.085°, 2418 independent (R_int_ = 0.0284) and 2393 observed reflections [I ≥ 2σ(I)], 194 refined parameters, R(F) = 0.0328, wR(F^2^) = 0.0905, max (min) residual electron density 0.531 (−0.300) e Å^–3^.

### 3.3. Synthesis

Starting materials 1,1′-ferrocenylenbis(*H*-phosphinic) acid **1** (H_2_fcd*H*p), 1,1′-ferrocenylenbis(metylphosphinic) acid **2** (H_2_fcd*Me*p), 1,1′-ferrocenylenbis(phenylphosphinic) acid **3** (H_2_fcd*Ph*p) were prepared according to procedure mentioned in the literature [46]. Their derivatives **1a** and **3a** [47], **2a** [48], **4a** and **4b** [51], **5a** [54], **5c** [54], **7** [54,58], **8a**–**d [54]**, **9a**,**b** [63], and Al or Fe aerogels [58,59] were prepared according to procedures mentioned in the literature. All the other chemicals and solvents in the synthesis were reagent grade and used as received.

*Synthesis of poly(1,1′-ferrocenediyl-bis(metylphosphinate) Zn(II))* (**6a**). Zn(NO_3_)_2_·6H_2_O (48 mg; 0.16 mmol) and H_2_fcd*Me*p (55 mg; 0.16 mmol) were dissolved in 10 mL of DMF/MeOH (1:2 *v*/*v*). The resulting mixture was heated for 12 h at 80 °C, a part of the methanol was evaporated and the orange powder of compound **6a** was obtained. Yield: 58 mg (96%) based on H_2_fcdMep. Anal. Calcd. for **6a** C_12_H_14_FeZnO_4_P_2_ (403.41 g/mol): C 35.70; H 2.97%. Found: C 36.37; H 3.09%. IR (nujol, cm^−1^): 3440 (w), 3088 (w), 1428 (w), 1372 (w), 1297 (m), 1228 (m), 1189 (s), 1152 (s), 1060 (s), 906 (m), 868 (m), 852 (m), 820 (m), 753 (s), 551 (w), 518 (m), 493 (m), 460 (m), 446 (m), 411 (m).

*Synthesis of poly(1,1′-ferrocenediyl-bis(metylphosphinate) Co(II))* (**6b**). Co(NO_3_)_2_·6H_2_O (47 mg; 0.16 mmol) and H_2_fcd*Me*p (55 mg; 0.16 mmol) were dissolved in 10 mL of DMF/MeOH (1:2 *v*/*v*). The resulting mixture was heated for 12 h at 80 °C, a part of the methanol was evaporated and the dark blue of compound **6b** was obtained. Yield: 56 mg (93%) based on H_2_fcdMep. Anal. Calcd. for **6b** C_12_H_14_FeCoO_4_P_2_ (396.95 g/mol): C 36.30; H 2.96%. Found: C 36.30; H 3.01%. IR (nujol, cm^−1^): 3441 (w), 1297 (w), 1219 (m), 1189 (m), 1139 (s), 1069 (s), 1058(s), 1049 (s), 905 (m), 868 (m), 852 (m), 820 (m), 762 (m), 753 (m), 550 (w), 518 (m), 493 (m), 463 (m), 446 (m), 413 (w).

*Synthesis of poly(1,1′-ferrocenediyl-bis(metylphosphinate) Mn(II))* (**6c**). MnCl_2_·4H_2_O (26 mg; 0.16 mmol) and H_2_fcd*Me*p (55 mg; 0.16 mmol) were dissolved in 10 ml of DMF/MeOH (1:3 *v*/*v*). The resulting mixture was heated for 12 h at 80 °C, a part of the methanol was evaporated and the orange crystal of compound **6c** was obtained. Yield: 58 mg (96%) based on H_2_fcdMep. Anal. Calcd. for **6c** C_12_H_14_FeMnO_4_P_2_ (394.96 g/mol): C 36.64; H 3.05%. Found: C 36.37; H 3.09%. IR (nujol, cm^−1^): 3426 (w), 3083 (w), 1428 (w), 1297 (w), 1212 (m), 1189 (m), 1144 (s), 1067 (s), 1050 (s), 905 (m), 867 (m), 852 (m), 821 (m), 752 (m), 542 (w), 512 (m), 493 (m), 459 (m), 444 (m), 408 (w).

*Synthesis of poly[Ni(N,N′-4,4′-bpy)(H_2_O)_4_]_2_(fcdMep) 2H_2_O* (**10**). Ni(NO_3_)_2_·6H_2_O (47 mg, 0.16 mmol), 4,4′-bipyridine (25 mg, 0.16 mmol) in 4 ml DMF/MeOH (1:1) solution was added to H_2_fcd*Me*p **2** (55 mg; 0.16 mmol) dissolved in DMF/MeOH (1:3 *v*/*v*, 4 mL). The resulting mixture was heated for 12 h at 80 °C, a part of the methanol was evaporated and the several orange crystals of compound was formed, which were suitable for X-ray diffraction analysis.

## 4. Conclusions

In this paper we summarized the synthesis of coordination compounds and MOFs based on ferrocenylbisphosphinic acids **1**–**3** with M(II) and M(III) ions. The results demonstrate that the type of acid and metal ion used significantly affect the resulting polymer’s structure and properties. Our study emphasizes the crucial role of ligand design in regulating the structure and properties of coordination polymers.

It was demonstrated that *H*-phosphinate plays a key role in variational mobility, while Me- or Ph-substituents of the phosphinate group can only lead to 0D complexes or 1D coordination polymers. With the use of additional ligands, such as 4,4′-bipyridine, the formation of a 3D structure is possible, but in this case, the use of *H*-phosphinate is necessary since the size of the substituent at the phosphorus atom has a significant effect on the formation of structures with different architectures. For acid **1**, a bridging type of coordination is observed due to the free rotation of the phosphinate groups around the P–C bond. Thus, this study highlights the importance of careful ligand design to control the structure and properties of coordination polymers.

However, further research is required to fully explore their potential in this area. The conclusions drawn in this paper will assist researchers in the design and synthesis of coordination polymers and MOFs in the future.

## Data Availability

The data presented in this study are contained within the article or are available upon request from the corresponding author Mikhail N. Khrizanforov.

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
