# Peer review of "Influence of the Substituent’s Size in the Phosphinate Group on the Conformational Possibilities of Ferrocenylbisphosphinic Acids in the Design of Coordination Polymers and Metal–Organic Frameworks"

_ijms, 2023, doi:10.3390/ijms241814087_

Round 1
Reviewer 1 Report
The work conducted by Khrizanforov and the co-workers summarized the modulation of conformational possibilities and coordination packing mode of the phosphinate group of ferrocenyl bisphosphinic acid-based coordination polymers by the size and type of the R substituent in the phosphinate group of the ferrocenyl bisphosphinic acids as well as the reaction conditions. For me, this is an interesting work. The manuscript was well written. I recommend its publication.
Minors:
1) P7, line 186, pare or pair?
2) Scheme 5, 6 and Figure 12 as well as the “materials and methods” part should be moved to the supporting information section.
Author Response
The work conducted by Khrizanforov and the co-workers summarized the modulation of conformational possibilities and coordination packing mode of the phosphinate group of ferrocenyl bisphosphinic acid-based coordination polymers by the size and type of the R substituent in the phosphinate group of the ferrocenyl bisphosphinic acids as well as the reaction conditions. For me, this is an interesting work. The manuscript was well written. I recommend its publication.
Minors:
1) P7, line 186, pare or pair?
2) Scheme 5, 6 and Figure 12 as well as the “materials and methods” part should be moved to the supporting information section.
Reply: Thank you very much for such positive comments. Corrected. We appreciate your time and effort in reviewing the manuscript.
Reviewer 2 Report
The manuscript "Influence of the Substituent's Size in the Phosphinate Group on the Conformational Possibilities of Ferrocenylbisphosphinic Acids in the Design of Coordination Polymers and MOFs" presents an interesting topic and provides fundamental insights into the influence of the substituent's size in the phosphinate group on the conformational possibilities of ferrocenylbisphosphinic acids in the design of coordination polymers and MOFs. The results presented in this manuscript are important for the design of phosphinate-tailored coordination polymers. However, the importance of phosphinate groups and their advantages in various applications are not discussed in this manuscript. Additionally, the results presented in this manuscript are not comprehensive and need to be improved. The authors should address these points before submitting the manuscript for publication
- Expand and clarify the figure captions throughout the manuscript.
- Provide a literature comparison table that shows how phosphinate groups have been used to improve the properties of coordination polymers and MOFs. (See, for example, ACS Omega 7 (18), 15275-152950.)
- Measure the N2 sorption isotherms and BET analysis of all the materials mentioned in the manuscript. (See, for example, Science Advances 8 (44), eade1473 and Journal of the American Chemical Society 145 (17), 9850-9856.)
- Revise the figures in the manuscript to improve their quality.
- Provide more detailed analytical characterizations of the materials presented in the manuscript. This should include PXRD compositions of pristine and activated material. (See, for example, these papers.)
- Provide a more thorough discussion of the applications and advantages of phosphinate-tailored coordination polymers.
- Conduct stability studies and discuss the advantages and potential of phosphinate groups for the design of coordination polymers and MOFs.
- Update the references in the manuscript to include more recent literature.
Author Response
The manuscript "Influence of the Substituent's Size in the Phosphinate Group on the Conformational Possibilities of Ferrocenylbisphosphinic Acids in the Design of Coordination Polymers and MOFs" presents an interesting topic and provides fundamental insights into the influence of the substituent's size in the phosphinate group on the conformational possibilities of ferrocenylbisphosphinic acids in the design of coordination polymers and MOFs. The results presented in this manuscript are important for the design of phosphinate-tailored coordination polymers. However, the importance of phosphinate groups and their advantages in various applications are not discussed in this manuscript. Additionally, the results presented in this manuscript are not comprehensive and need to be improved. The authors should address these points before submitting the manuscript for publication
Reply: We appreciate your time and effort in reviewing the manuscript. Thank you for your comment. The description has been updated. The main point of this work is that the substituent's size in the phosphinate group influences the conformational possibilities of ferrocenylbisphosphinic acids.
Q1: Expand and clarify the figure captions throughout the manuscript.
Reply: Corrected.
Q2: Provide a literature comparison table that shows how phosphinate groups have been used to improve the properties of coordination polymers and MOFs. (See, for example, ACS Omega 7 (18), 15275-152950.)
Provide a more thorough discussion of the applications and advantages of phosphinate-tailored coordination polymers. Conduct stability studies and discuss the advantages and potential of phosphinate groups for the design of coordination polymers and MOFs.
Reply: We have added a comparison between carboxylates and phosphinates in the introduction based on the study in the review. However, our work does not aim to compare all phosphinates, and therefore, a comparison table would be redundant. The advantages and potential of phosphinate groups for the design of coordination polymers and MOFs are well described in the review "Phosphinic acids as building units in materials chemistry" by Kloda, M. et all, Coord. Chem. Rev. 2021, 433, 213748. To avoid repetition, we have cited this work and highlighted the main points in the manuscript (highlighted in yellow).
Q3: Measure the N2 sorption isotherms and BET analysis of all the materials mentioned in the manuscript. (See, for example, Science Advances 8 (44), eade1473 and Journal of the American Chemical Society 145 (17), 9850-9856.
Provide more detailed analytical characterizations of the materials presented in the manuscript. This should include PXRD compositions of pristine and activated material. (See, for example, these papers.)
Reply: Thank you for your question. For porous polymers, we have published nitrogen sorption isotherms in the corresponding articles.
Shekurov, R.; Miluykov, V.; Kataeva, O.; Krivolapov, D.; Sinyashin, O.; Gerasimova, T.; Katsyuba, S.; Kovalenko, V.; Krupskaya, Y.; Kataev, V.; Büchner, B.; Senkovska, I.; Kaskel, S. Reversible water-induced structural and magnetic transformations and selective water adsorption properties of poly (manganese 1,1′-ferrocenediyl-bis (H-phosphinate)). Cryst. Growth Des. 2016, 16 (9), 5084–5090.
Shekurov, R.P.; Gilmanova, L.H.; Miluykov V.A. New porous Fe(III)-based ferrocene-containing diphosphinate. Phosphorus, Sulfur, Silicon Relat. Elem. 2019, 194 (10), 1007–1009.
Khrizanforova, V.V.; Shekurov, R.P.; Nizameev, I.R.; Gerasimova, T.P.; Khrizanforov, M.N.; Bezkishko, I.A.; Miluykov, V.A.; Budnikova Y.H. Aerogel based on nanoporous aluminium ferrocenyl diphosphinate metal-organic framework. Inorganica Chim. Acta. 2021, 518, 120240.
Khrizanforova, V.; Shekurov, R.; Miluykov, V.; Khrizanforov, M.; Bon, V.; Kaskel, S.; Gubaidullin, A.; Sinyashin O.; Budnikova, Y. 3D Ni and Co Redox-Active Metal-Organic Frameworks Based on Ferrocenyl Diphosphinate and 4,4’-Bipyridine Ligands as Efficient Electrocatalysts for Hydrogen Evolution Reaction. Dalton Trans. 2020, 49, 2794–2802.
Compound 7 is a discrete molecule, and its porosity was not measured. According to TGA and X-ray analysis, there are no additional solvent molecules for 1D polymers 4 and 6, as well as 2D polymer 8. TGA for these samples demonstrates stability at high temperatures, indicating the absence of porosity in the polymers. Otherwise, weight loss due to the loss of solvent molecules would have been observed. In this regard, only compounds 5 and 9, as well as aerogels of Fe(III) and Al(III), are porous.
For compound 5 - a 2D polymer, it demonstrates a reversible crystal-to-amorphous transformation (Figure 1). The activated form does not sorb nitrogen and selectively sorbs water. Please refer to the article on manganese for more information. Upon heating up to 150°C, both coordinated and lattice water molecules are eliminated, producing a compound that is non-porous for nitrogen but can selectively adsorb water over methanol and other solvents at 298 K.
Figure 1. Powder XRD patterns: a - simulated from the single-crystal X-ray data of 5; b - as-synthesized 5; c - the dehydrated form of 5 (compound 5'); d - the rehydrated 5’.
The reversible structural transformation during adsorption/desorption of water is also reflected in a change of magnetic properties of the MOF. The porosity of the compound after dehydration was investigated by nitrogen physisorption at 77 K, and it was shown, that the dehydrated form of 5 is not accessible for nitrogen molecules. In contrast, the compound adsorbs water vapor at 298 K. Starting from low pressure, a slowly growing uptake can be observed up to p/p0 0.8, followed by steep rise in the adsorbed amount at relative pressures above 0.8 (see. Figure). Such profile of adsorption isotherm can be explained by the flexible behavior of the network during water sorption. The Type II adsorption isotherm transforms to Type I which indicates the structural rearrangement of the MOF at “gate opening” pressure. The uptake in saturation corresponds well to the four H2O molecules per formula unit and 17% weight loss observed from TG curve. Thus, the activated compound is able to adsorb water selectively over other solvents. The desorption branch of the isotherm does not fit the adsorption branch, forming a broad hysteresis, which is also characteristic for “gate-pressure” MOFs. Obviously, the incorporated water molecules cannot be easily desorbed even at low relative pressure. To desorb water molecules, heating is needed in addition to evacuation. Moreover, one should note that the desorption process of 5 significantly differs from that observed in other flexible porous coordination polymers containing both coordinated and crystal lattice water molecules. Many of them start to lose lattice water already at ambient conditions, the others exhibit step-like water desorption curves, while 5 is extremely stable at room temperature and loses all 4 water molecules in one step upon heating.
The water adsorption/water removal process is completely reversible and can be repeated. In the second water vapor adsorption cycle, after sample evacuation at 150 °C over night, the amount of water adsorbed as well as the shape of adsorption isotherm remain the same (Figure 2).
Figure 2. Adsorption (closed symbols) and desorption (open symbols) isotherms of H2O vapor (298 K) (black circles - cycle 1, and red squares – cycle 2), and MeOH (298 K) (blue triangles) for 5’.
The aerogels (based on AlIII and FeIII) exhibit a N2-physisorption isotherm of type III, which is indicative of macroporous solids. No hysteresis is observed between the desorption and adsorption isotherms of the aerogel, indicating that there is little irreversible interaction between N2 and the pores. The aerogel may thus exhibit a colloidal microstructure rather than a polymeric one. Additionally, the aerogel exhibits high surface areas of approximately 671 m2/g.
For 3D MOF 9 - TGA, high stability and the absence of a plateau of a stable form during the desorption of the polymer were demonstrated. Nitrogen sorption for dehydrated forms after 150 or 250 degrees did not show significant results. The sample loses sorbed molecules with irreversible destruction of the crystalline lattice.(Figures 3-4)
Figure 3. Adsorption (closed symbols) and desorption (open symbols) isotherms of N2 for 9’.
Figure 4. Thermogravimetric analysis of 3D coordination polymers 9a (black curve) and 9b (gray curve).
For the new crystals presented in this article, such as 6, there is no PXRD of the activated variant. The TGA for polymer 6 demonstrates stability up to 400 ⸰C, indicating the absence of porosity in the polymer. Otherwise, we would observe weight loss with the loss of solvent molecules.
Q4: Revise the figures in the manuscript to improve their quality.
Reply: Corrected.
Q8: Update the references in the manuscript to include more recent literature.
Reply: Corrected.

Round 2
Reviewer 2 Report
I thank the authors for revising the manuscript and incorporating the suggested changes. Despite our recommendations to enhance the manuscript's quality, the authors have not executed the requisite experiments as advised. Notably, including N2 sorption isotherms and BET analysis is imperative, mainly when dealing with novel MOF materials. It is essential to emphasize that the absence of porosity analysis, specifically the omission of N2 sorption isotherms and BET analysis, might necessitate the consideration of a more suitable journal for submission, such as "Crystals MDPI." The authors are commended for their efforts in addressing previous concerns; however, there remains a need for significant revisions in line with the outlined suggestions to meet the standards expected by the IJMS journal.
The following suggestions must be thoroughly integrated into the revised manuscript:
- The manuscript should encompass porosity analysis and BET surface area evaluation, as initially proposed during the first revision (refer to point 3 of my previous suggestions in revision 1).
- The main manuscript should present A comprehensive PXRD analysis (see point 4 of my prior revision recommendations in revision 1).
- Addressing stability studies, while TGA analysis contributes to the initial understanding of MOF's thermal stability, it is imperative to explore its chemical stability encompassing solvents, acids, and bases to facilitate potential applications (as referenced in Science Advances 8 (44), eade1473). This journal seeks not only to showcase novel MOFs but also to highlight their practical applications and advantages. Consequently, the presentation of TGA and PXRD results within the main manuscript is warranted to align with the standards of the IJMS journal.
- The assertion made by the authors regarding the reversible crystal-to-amorphous transformation of compound 5, a 2D polymer, lacks sufficient substantiation beyond the provided crystallographic image (Figure 1). To address this, I recommend the inclusion of systematic structural transformation studies supported by PXRD analysis to elucidate any structural changes.
- Notably absent from the manuscript are the N2 sorption data, despite the authors referencing figures 2, 3, and 4 in their response. It is essential that the presented data in the manuscript appropriately reflect the reviewers' inquiries and expectations.
- Refer to the below publications for more details as suggested in the 1st revision (See, for example, ACS Omega 7 (18), 15275-152950.See, for example, Science Advances 8 (44), eade1473 and Journal of the American Chemical Society 145 (17), 9850-9856)
Author Response
Thank you for your detailed feedback and recommendations regarding our manuscript. We acknowledge the significance of N2 sorption isotherms and BET analysis, especially in the context of novel MOFs. However, it's important to clarify our perspective on this. Requesting sorption data for non-porous materials, in our view, can be likened to asking for seeking enantioselective catalysis data for a non-chiral catalyst. That said, we've tried to incorporate most of the suggested experiments and have included them in the Supporting Information for reference.
Originally, our intention with this work was to primarily highlight the structural characteristics. Recognizing that this focus might be narrow, we expanded the manuscript to include a section (2.4. Physico-chemical properties.) detailing the applications of the synthesized polymers. Furthermore, for the previously unreported polymers, we have provided their electrochemical properties. We truly appreciate your time and effort in reviewing our work and hope that these clarifications and additions address the concerns adequately.
